# Mysterious Morphology: An Investigation of the Octopus Keel and Its Association with Burrowing

**DOI:** 10.3390/biology12091204

**Published:** 2023-09-04

**Authors:** Cheyne Springbett, Katie Cordero, Garrett Ellis, Carly Haeger, Kirt L. Onthank

**Affiliations:** Department of Biological Sciences, Walla Walla University, College Place, WA 99324, USA

**Keywords:** octopus, burrowing, keel, *Muusoctopus leioderma*, sediment, phylogeny, evolution

## Abstract

**Simple Summary:**

Octopuses are a diverse group of charismatic animals capable of adapting to a wide range of environments. Marine sediment habitats are the most pervasive environments on Earth and are largely dominated by burrowing organisms. Burrowing, the formation of semi-permanent structures below the surface of the sediment bed, is a novel behavior among octopuses, and the morphology facilitating burrowing in most octopuses is unknown. The goal of this study was to investigate the octopus keel, a fold of skin that protrudes from the lateral margin of the mantle in some species, as a burrowing-associated trait. The keel has been noted in several octopuses known to burrow and has been hypothesized to be associated with burrowing. We found that burrowing octopus species were more likely to also have keels, and that burrowing species of octopus, when held in aquaria, were more likely to lose their keels if they did not burrow. This article represents the first phylogenetic evidence of a connection between keels and burrowing, as well as evidence of the degeneration of keels in non-burrowing *Muusoctopus leioderma*.

**Abstract:**

The octopus keel is a trait that has been hypothesized to be connected with burrowing in octopuses, but has never been explored in any detail. We investigated the association between these two traits using two approaches. First, we examined the phylogenetic correlation between the presence of a keel and known burrowing behavior in cirrate octopuses. Second, burrowing and non-burrowing captive *Muusoctopus leioderma* were evaluated for keel prominence to determine whether the keel is lost more rapidly in non-burrowing individuals. Pagel’s test for the coevolution of binary characteristics showed the model of best fit for the resulting phylogenetic tree to be one of evolutionary interdependence, and that non-burrowing *Muusoctopus leioderma* lost their keels over time, while burrowing individuals maintained their keels. Together, these results indicate the keel may be a trait associated with burrowing in octopuses.

## 1. Introduction

Marine sediments are the most expansive and pervasive environments on Earth [1]. These seemingly barren underwater habitats make up roughly 70% of Earth’s surface, yet their depths and temperatures make them the least accessible environments possible for humans [1]. These habitats host a diverse range of marine life, and are largely dominated by burrowing organisms. Little is known about the behavior and ecology of these marine infauna, as the difficulty of examining these habitats has stymied their research [1].

Octopuses are a diverse group of invertebrates found in most marine environments. Charismatic, highly intelligent, and displaying a wide range of novel adaptations for invertebrates, this group plays an important role, both in ecological systems and in facilitating public engagement and interest in science [2]. Octopuses act as both generalist predators and prey, playing crucial roles in trophic webs and ecological health [3,4,5]. Cephalopods, including octopuses, are also expanding in range and population [5,6,7,8], suggesting that their ecological significance will grow and sparking questions about octopus’ resilience and adaptation.

The smooth-skinned octopus (*Muusoctopus leioderma)* is a deep-water species and most commonly found between 450 and 650 m [9] throughout the northern Pacific Ocean, from California up to the Sea of Okhotsk, off Siberia, and until recently had never been recorded at depths shallower than 70 m [9,10]. However, a population of *M. leioderma* has recently been found to reside in Burrows Bay, WA, USA, at only 15 m depth and accessible by SCUBA [11]. This exciting discovery allows for the collection and study of live and minimally disturbed benthic octopuses via SCUBA that would normally be inaccessible.

This species actively modifies its environment by burrowing, an adaptation allowing organisms to move through and compact soft substrata to form semi-permanent structures [12]. Burrowing organisms have an unusual level of influence on their environment, by directly altering marine sediment biochemistry and structure [13] and indirectly by impacting co-occurring bacteria, microalgae, macrofauna, seagrasses, and other secondary consumers affected by sediment nutrition [13].

*Muusoctopus leioderma*’s most notable defining characteristic is the presence of a keel, a flap of skin that protrudes from the lateral margin of the mantle [11]. This trait, shared with some other species of octopuses, has been hypothesized to be linked to burrowing in several benthic species [14]. The keel is essentially just an expandable flap of skin, and lacks rigidity, and is therefore unlikely to be used directly for excavation. However, the keel’s apparent connection with known burrowing octopus species makes it an intriguing point of study as a possible burrowing trait for benthic octopuses and may lend clues to the burrowing strategy of *M. leioderma* and other octopuses.

## 2. Materials and Methods

### 2.1. Octopus Collection

Octopuses (*n* = 10) were collected from Burrows Bay, Skagit County, WA, USA (48°28′12″ N, 122°40′53″ W) by SCUBA. Once found, they were captured and returned to Rosario Beach Marine Laboratory (RBML) in re-sealable plastic storage bags filled with saltwater. Upon arrival at RBML, octopuses were sexed, weighed, and assigned a name. Lengths were also obtained in the laboratory, using the same laser dot camera method used to measure lengths during BRI recordings in situ (see below). These data, along with species, date caught, depth, the tank number the octopuses were to be placed in, and any notes, were recorded. Then, octopuses were immediately transferred to a 61 cm × 33 cm × 41 cm acrylic aquarium, or “mud tank”, filled approximately halfway with sediment from Burrows Bay, for their 24 h laboratory acclimation period. The time that octopuses began this acclimation period, along with the date, sex, name, and tank, were recorded in Google Sheets. A total of 10 individuals were collected and were observed for keel prominence.

### 2.2. Octopus Holding

Octopuses were held in aquaria filled with sediment collected from Burrows Bay. These “mud tanks” were connected to RBML’s seawater system to provide fresh saltwater. Tanks were modified for a flow-through system, with an inflow and passive outflow (Figure 1). Tanks’ outflow lines ran to a single plastic tub with a sump pump connected to a float switch, allowing all tanks to have short lines and consistent outflow to a single drain. Laboratory lighting conditions were controlled and matched with natural lighting conditions in Burrows Bay using custom lights. These lights were diffused green LEDs, controlled by a Raspberry Pi Pico microcontroller, and powered by a DC wall socket. Each mud tank was equipped with a light array programmed to brighten and dim closely following a diurnal cycle, approximating what occurs naturally. Peak brightness hours and ramp-down periods were calculated using the “sunriset” function from the maptools R package [15] using 31 July 2022, a date roughly halfway through our data collection period, as a set date, and Burrows Bay, Skagit County, Washington, USA, as our set location. There were also red-light sources that were constantly illuminated and allowed nighttime photography of octopuses when the green lights were off without disturbing them. *Muusoctopus leioderma*’s eyes are non-reactive (no discernable pupillary response) to red light (pers obs.).

Sediment for the tanks was collected directly from Burrows Bay either by SCUBA or using a box core sampler or Van Veen grab samplers. Sediment was collected from the surface of the sediment bed in Burrows Bay and was measured using standard wet-sieving by lightly running water over sediment and shaking sediment on scientific grade sieves until the water ran clear. Octopuses were held for at least 12 days, then periodically released by divers while new individuals were collected. Octopuses ate benthic infauna, primarily polychaete worms, present in the collected sediment.

### 2.3. Muusoctopus leioderma Keel Prominence

After their initial acclimation period, captured individuals were periodically evaluated for keel prominence by taking pictures of the octopus’s keel with an iPhone 12 mini. Target keel days were days 2, 6, and 12 in captivity. However, keels could not be photographed when the octopus was in its burrow, so occasionally octopuses would have to be photographed later, as soon as the octopus was visible. Pictures were taken opportunistically on target dates when octopuses had voluntarily left their burrows. This was necessary to avoid forcibly removing them and impacting the study. Pictures were taken during nighttime hours between 21:00 and 09:00, since this was the most likely time that they would be visible and out of their burrows. Octopuses were kept in treatment for a target of 3 keel collection dates in the mud tanks, although time restraints of the field season and laboratory availability forced treatment to end early in some cases. After completing their treatment in the mud tanks, individuals were transferred to non-light-controlled plastic containers, or “Octocondos”, connected to RBML’s saltwater system, to await return and release to Burrows Bay via SCUBA, and allowing new octopuses to begin treatment. In some cases, the octopuses never emerged from their burrow after their initial burrowing event. These octopuses were dug out of their burrows after 10–20 days. Occasionally, such as when octopuses needed to be dug out from their burrows, images for keel evaluations were collected after octopuses were transferred to the Octocondos, and this was noted in the data. Later, individuals were categorized into two groups: burrowers, or octopuses that burrowed below the sediment layer, and non-burrowers, or octopuses that did not burrow in captivity.

Keels were evaluated qualitatively based on 5 levels of prominence, with a 0 or “Missing” being a completely absent keel, and a 4 or “Very Prominent” being the strongest level of prominence, using a key created for this purpose (Appendix A). The key provided descriptions of distinctions between each of the levels of prominence, as well as multiple images highlighting those differences. Keels were evaluated independently by 4 observers, and the consistency of the evaluations was examined. These were blind evaluations, meaning evaluators did not know if images were of burrowing or non-burrowing octopuses, and they lacked access to other evaluators’ results.

Each observer had their own private sheet to perform evaluations independently from other observers. These results were averaged to obtain keel prominence values for each keel measurement of all individual octopuses. Individual octopuses were split into two groups based on burrowing status: burrowers and non-burrowers. Keel evaluations were categorized as either mud tank or Octocondo evaluations. Individuals that had one or fewer keel measurements were excluded because keel loss could not be determined. Octopuses’ keel prominence was compared with how many days they had been held in captivity, which was calculated using the dates of octopus collections and keel evaluations.

To test consistency between evaluators, we calculated an intraclass correlation coefficient (ICC) as an index of interrater reliability. ICC was run as a two-way consistency model using the irr package in R [16].

Statistically significant differences in keel prominence between burrowing and non-burrowing octopuses were analyzed using a Linear Mixed Effect (LME) model from the nlme package in R [17]. A total of 10 individuals were collected. One octopus escaped its tank, and data was lost for one octopus, so eight individuals (n = 8) were used for analysis. The LME model accounted for two different keel measurement scenarios: mud tanks or Octocondos. This model was analyzed using an ANOVA test from the car R package [18].

### 2.4. Octopoda Keel and Phylogenetics

Phylogenetic analysis of the occurrence of keels across benthic octopuses in the order *Octopoda* in relation to burrowing was accomplished using visual identification of keels as “Present” or “Absent” by examining images from the literature, species descriptions, and by reaching out to species-specific experts. These species were classified as “known to bury or burrow” or “not known to bury or burrow”. The octopus species selected for analysis were all benthic species with known genetic sequencing of the cytochrome oxidase subunit 1 (COI), cytochrome oxidase subunit 3 (COIII), and/or 16S genes obtained from GenBank. These genes were chosen because they provided the best coverage of the greatest number of species, and each of these species had sequences of one or more of these genes available, sufficient for an alignment. Species lacking genetic sequencing, or that lacked data on both keels and burrowing behavior, were mostly deep-water species with very little research focus, and these were omitted from the analysis. The final tree included 110 total species. For octopuses in which only keel presence was unknown, the keel was considered missing. Octopuses in which only burrowing behavior was unknown were considered “not known to bury or burrow”. Evolutionary correlation between mantle keel and burrowing or burying behavior were tested using Pagel’s test in R.

Genetic sequences were obtained from GenBank using the read.GenBank function from the ape R package using accession numbers [19] for benthic octopus species with known sequencing of COI, COIII, and 16S, as these 3 genes provided the best genetic coverage of benthic octopuses. Sequences were aligned using the AlignSeqs function from the DECIPHER package [20]. This allowed for the creation of a NEXUS file that could then be used in creating the final tree. For each of the three genes, a model test was conducted with the phangorn package for each codon position independently, so the dataset could be partitioned into each of the three codon positions [21].

The resulting files were combined into a single nexus file and a MrBayes command block was generated using a custom BASH script. This file was then used for the final analysis and production of the multi-gene tree using MrBayes [22].

To test for a statistically significant evolutionary relationship between keel presence/absence and known burrowing behavior, the multi-gene tree was analyzed using Pagel’s test. Pagel’s test examines the independent evolution of 2 binary characters by comparing a ratio of likelihoods of two models [23]. Pagel’s test was run using the phytools R package [24].

## 3. Results

### 3.1. Keel Regression in Non-Burrowing Muusoctopus leioderma

Keel evaluations of captive octopuses showed a clear effect of burrowing on keel prominence. Keel evaluations were conducted blindly and independently by four observers, and the results were consistent (ICC = 0.91, *p*-value < 0.001, 95% confidence interval: 0.85 < ICC < 0.95). Average keel prominence values of the linear mixed effects model with an ANOVA analysis showed a statistically significant interaction between days and burrowing status, meaning that the condition of an individual being a burrower vs. a non-burrower had a significant effect on the relationship between days in captivity and keel prominence. (Table 1, Χ^2^ = 4.66, df = 1, *p*-value = 0.031). Specifically, we saw non-burrowing individuals exhibiting a negative relationship between keel prominence and time in captivity, while burrowing individuals showed no such pattern of keel loss (Figure 2). There was also a significant effect of day on keel prominence (Table 1, Χ^2^ = 4.37, df = 1, *p*-value = 0.037). In addition, the analysis found that whether the image was taken in the mud tank or in the Octocondo after removal from the mud tank did not have a significant effect on these results (Table 1, Χ^2^ = 0.84, *p*-value = 0.358). The sediment in the mud tanks, collected from Burrows Bay, was measured as <220 um, or a fine, silty type sediment.

### 3.2. Keel Evolution in Burrowing Octopoda

The phylogenetic tree, when analyzed using Pagel’s test, produced a matrix of the phylogenetic relationships between four conditions: keel present, keel absent, known burrowing, and no known burrowing (Figure 3). Pagel’s test produced a statistically significant result by testing four models of evolutionary pressure, and found the model of best fit to be one of interdependence between the presence of a keel and known burrowing, demonstrating evidence of an interdependent evolutionary relationship between these phylogenies in benthic octopuses (Table 2, AIC = 188.43, *p*-value < 0.001). Additionally, the model testing independence, or the absence of an evolutionary relationship between these characters, was the worst fitting and least likely model tested (Table 2, AIC = 200.98).

## 4. Discussion

In burrowing captive *Muusoctopus leioderma*, we saw a significant interaction between the effect of octopus burrower/non-burrower status and number of days in the mud tank on the keel prominence, meaning that the condition of an octopus being a burrower or non-burrower impacted how keel prominence changed over time. Specifically, we saw a stronger negative relationship between keel prominence and time in captivity for non-burrowers than for burrowers (Figure 2). This suggests that the keel is better maintained when octopuses continue burrowing in captivity. The mechanism behind this is still unclear, as the precise function of the keel is still unknown. It is possible that the keel is a “use it or lose it” trait, and that the keel disappears when the octopus does not continue the behavior. Keel loss could also be a result of stress from captivity, with burrowing individuals being less prone to stress in their tanks, despite the minimal disturbance and light-controlled nature of the experiment.

Then, we broadened our focus to benthic octopuses as a whole to explore the possibility of an evolutionary relationship between known burrowing behavior and keel presence. Pagel’s test is a statistical test that compares models of evolutionary relationships between binary traits, and this test revealed that the model of best fit for the phylogenetic tree was one of interdependence, suggesting the two traits co-evolved. While keels have been anecdotally linked to burrowing in octopuses [14], this is the first study to provide phylogenetic evidence of a link between the two traits. It was originally speculated that the keel may act as a kind of rudder, allowing the octopus to orient and guide itself into and/or through the sediment [14]. Octopus keels are essentially a flap of skin with very little rigidity or structure, so it is less likely that it functions directly as an excavating appendage to displace sediment. However, the extra skin is the only external textural feature in this species, as *M. leioderma* lacks any papillae or other prominent skin texture. This unusually smooth and streamlined profile is ideal for burrowing, as it may help to reduce resistance from marine sediment. The keel is the only thing that disrupts this low resistance body shape, but the notion that this skin is being used to displace sediment is unlikely.

Instead, the keel may be involved in improving the efficiency of burrowing. In general, there are advantages for burrowing organisms with smaller body sizes. Smaller animals are faster and expend less energy in making their burrows compared to larger organisms [25,26]. However, there are also advantages to constructing a larger burrow. For example, a larger burrow would allow for improved water flow and could reduce the risk of anoxia in such a confined environment. The width of a burrow is usually proportional to the size of the organism [27], but larger individuals burrow more slowly [28] and expend more energy than smaller individuals [25,26]. A keel may help to offset these issues while allowing the octopus to build a larger burrow.

To accomplish this, the keel may act as an extra reservoir of skin that allows the octopus to expand its mantle during the burrowing process. Octopuses expand and retract their mantles as part of the normal respiration process, and octopus skin is very flexible, normally retracting into a highly textured surface of papillae or folds [29]. *Muusoctopus leioderma*’s skin texture is concentrated at the lateral margin of the mantle, but the mantle still expands and retracts during respiration. It is possible that the extra skin allows the octopus to expand its body size while burrowing beyond what would be possible otherwise, allowing for a disproportionately large burrow, and overall less resistive skin for burrowing. Additionally, mantle expansion may also be a part of this octopus’s overall burrowing strategy. The razor clam (*Ensis directus*) utilizes a strategy of bodily expansion and retraction in its burrowing, and it is possible that *M. leioderma* could be burrowing in a similar way [30].

The elastic nature of octopus skin may also explain the loss of keels in captive non-burrowers. Individuals no longer expanding their mantles during burrowing could show reduced keels as the extra skin is pulled back into the mantle from disuse. There is currently no research on the impacts of inactivity on octopus skin, but inactivity in humans results in reduced skin elasticity, muscle mass, and tone, and alterations to the skin and muscle fiber composition [31].

Alternatively, the keel may just be an adaptation connected with *M. leioderma*’s smooth skin to improve sediment resistance. We measured the sediment in Burrows Bay, and found it to be a fine, silty sediment. Burrowing strategies are largely dictated by the type of sediment the organism burrows in [12], so gaining a better idea of sediment composition in Burrows Bay may be helpful in future interspecific studies of burrowing mechanics and the keel’s function. Next steps would involve trying to see *M. leioderma* burrowing in a transparent medium in the laboratory to document the keel below the sediment, and to investigate interspecific variation in mantle plasticity and skin texture to explore connections between octopus skin and burrowing behavior, as well as recording the burrowing process with a high frame rate camera in keeled octopuses.

## 5. Conclusions

Burrowing is a life-strategy closely linked with marine sediment and deep-sea environments, and this connection makes burrowing an ideal target for understanding these inaccessible and understudied ecosystems. The evolution of phenotypes that facilitate this behavior varies based on both organism and environment. This study found evidence that the keel may be a burrowing-associated trait in octopuses, using two independent sets of methods. Phylogenetic analysis of known burrowing and keel presence in benthic octopuses provided a model of best fit of evolutionary interdependence between keels and known burrowing. Meanwhile, individual non-burrowing *Muusoctopus leioderma* were shown to lose their keel faster compared with burrowing individuals. Both of these results are what we would expect to see if this were a burrowing-associated trait, and together indicate that this is a phenotype linked with burrowing in octopuses.

## Figures and Tables

**Figure 1 biology-12-01204-f001:**
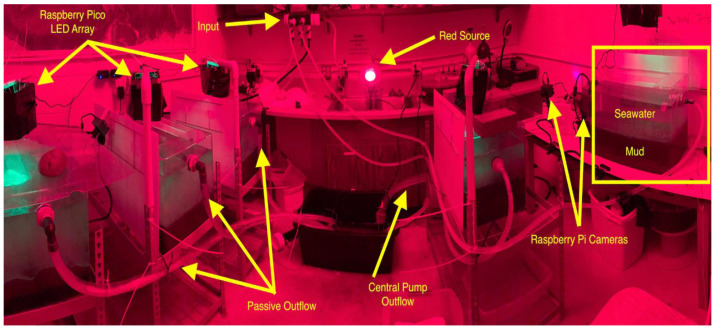
Octopus holding and laboratory setup, with water system, lighting, and all 5 experimental mud tanks shown.

**Figure 2 biology-12-01204-f002:**
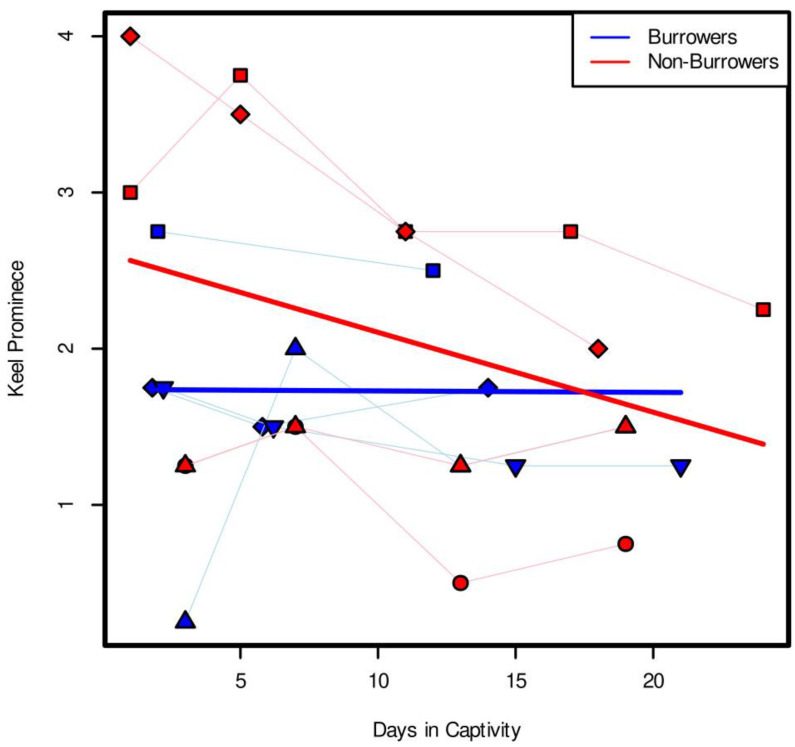
Keel prominence values vs. time in captivity for burrowing and non-burrowing individuals. Individual shapes connected by thin lines show keel prominence for individual octopuses during captivity, with blue showing burrowing individuals and red showing non-burrowers. Thick lines show results of LME. Results show a significant interaction between burrowing and day, with non-burrowing octopuses losing their keels faster than non-burrowers (ANOVA, days/burrow, Χ^2^ = 4.66, df = 1, *p*-value = 0.031).

**Figure 3 biology-12-01204-f003:**
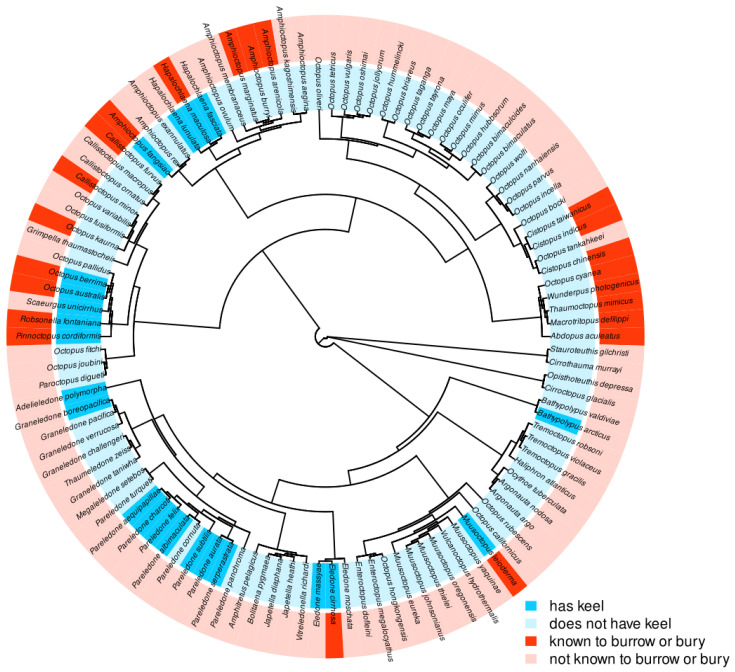
Multi-species phylogenetic tree showing results of Pagel’s binary character correlation test between keel presence and known burrowing/burying behavior. Tree includes 110 benthic octopus species. Results show interdependence to be the model of best fit (AIC = 188.43, *p*-value < 0.001).

**Table 1 biology-12-01204-t001:** Analysis of deviance table, summarizing ANOVA test of variables impacting keel regression. Statistically significant results, including the interaction between days and burrow, are denoted with an asterisk (*).

	Χ^2^	Df	*p*-Value
days	4.37	1	0.037 *
burrow	2.09	1	0.149
type	0.84	1	0.358
days/burrow	4.66	1	0.031 *

**Table 2 biology-12-01204-t002:** Results of Pagel’s test from the ARD substitution model used between dependent and independent model rate matrices with AIC values for the four evolutionary relationships tested: burrowing, non-burrowing, keel, no keel. *p*-value < 0.001.

Model	AIC
keel-dependent	190.56
burrow-dependent	197.54
interdependent	188.43
independent	200.98

## Data Availability

The codes for *Muusoctopus leioderma* keel evaluations presented in this study are openly available in Zenodo at https://doi.org/10.5281/zenodo.8021840 (accessed on 1 January 2023). The codes for octopus keel phylogenetic evaluations and accession numbers used in this study are openly available in Zenodo at https://doi.org/10.5281/zenodo.8021847 (accessed on 1 January 2023). Keel evaluation images used in this study are openly available in Zenodo at https://doi.org/10.5281/zenodo.8040502 (accessed on 1 January 2023).

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
