# Peer review of "Mysterious Morphology: An Investigation of the Octopus Keel and Its Association with Burrowing"

_biology, 2023, doi:10.3390/biology12091204_

Round 1

Reviewer 1 Report

Please improve a clarity of your Figures. My comments are on the attached file

Author Response

Thank you very much for your input! Please find our responses in the attached file.

Reviewer 2 Report

Nice short manuscript. There are several minor suggestions for Authors to improve the clarity of their presentation:

-why figures are in separate chapter (3.3)? Place them in the text right after they are cited;

-statistical tests and p-values have up to 9 (!) decimals... 2 is enough for everything but significant p-values, where 3 or 4 is enough;

-lines 186 to 191: provide full models' data in a table;

--lines 196 to 197: another table with these models.

Author Response

(The authors gave the same response as above.)

Reviewer 3 Report

All Figures should be placed in the subsections where they are mentioned, i.e. Figure 1 under subsection 2.2.; Figure 2 under subsection 3.1.; Figure 3 under 3.2…

Line 73: How many animals were collected? It is not quite understandable how many animals were in the experiment. How many were used for statistical analysis?

Line 84: In subsection 2.2 Octopus holding, it should be explained whether the animals were fed, when and with what food.

Line 109: What do you mean by “normally”? If you are conducting an experiment, why would randomly take pictures? It is not clear from the MS whether observers watched videos or took pictures.

Line 110: “Octopuses were kept in treatment for a target of 3 keel collection dates in the mud tanks, although time restraints forced treatment to end early in some cases.“ Could you explain these time restraints? Why did some treatments end earlier? 

Line 179: Results are really poorly written. This part should be precisely explained and written in a sense to make it easier for the reader to follow the study and its findings. Describe what the statistical significance was, positive, or negative...

Line 230: Keel is „use it or lose it„ trait? Do you mean this in evolutionary terms? From the sentence, it would appear that this is stated on an individual level.

Line 296 – 298. „Figure 1: Octopus holding and laboratory setup; Figure 2: Keel prominence 296 values vs time in captivity for burrowing and non-burrowing individuals; Figure 3: Multi-species 297 phylogenetic tree showing keel presence and known burrowing.“ Figures are not part of the Supplement material, but part of the manuscript.

Line 321: Appendix A: Keel Evaluation Key should be used as Supplementary material. All the photos lack pointers to the keel. Also, why are there no photos of stages M – Missing and W – Weak? All the given stages are not clearly recognizable, the keel seems more or less the same. It did not convince me that there were this many differences. According to the presented images, there is no clear difference between these stages.

Author Response

(The authors gave the same response as above.)

Round 2

Reviewer 3 Report

My concern is the live animals' welfare in this study. 

The experimental design is not good and it is not strongly presented.

Presented „Keel Evaluation Key“ did not provide clear recognition of different keel stages.

Introduction lacks clear aims and hypotheses.

Results are poorly written, while Discussion is more like an extended introduction, with a lot of information repeated.